# Impact of Lignocellulose Pretreatment By-Products on *S. cerevisiae* Strain Ethanol Red Metabolism during Aerobic and An-aerobic Growth

**DOI:** 10.3390/molecules26040806

**Published:** 2021-02-04

**Authors:** Grzegorz Kłosowski, Dawid Mikulski

**Affiliations:** Department of Biotechnology, Kazimierz Wielki University, ul. Poniatowskiego 12, 85-671 Bydgoszcz, Poland; klosowski@ukw.edu.pl

**Keywords:** biomass pretreatment by-products, yeast stress, *S. cerevisiae*, HSP

## Abstract

Understanding the specific response of yeast cells to environmental stress factors is the starting point for selecting the conditions of adaptive culture in order to obtain a yeast line with increased resistance to a given stress factor. The aim of the study was to evaluate the specific cellular response of *Saccharomyces cerevisiae* strain Ethanol Red to stress caused by toxic by-products generated during the pretreatment of lignocellulose, such as levulinic acid, 5-hydroxymethylfurfural, furfural, ferulic acid, syringaldehyde and vanillin. The presence of 5-hydroxymethylfurfural at the highest analyzed concentration (5704.8 ± 249.3 mg/L) under aerobic conditions induced the overproduction of ergosterol and trehalose. On the other hand, under anaerobic conditions (during the alcoholic fermentation), a decrease in the biosynthesis of these environmental stress indicators was observed. The tested yeast strain was able to completely metabolize 5-hydroxymethylfurfural, furfural, syringaldehyde and vanillin, both under aerobic and anaerobic conditions. Yeast cells reacted to the presence of furan aldehydes by overproducing Hsp60 involved in the control of intracellular protein folding. The results may be helpful in optimizing the process parameters of second-generation ethanol production, in order to reduce the formation and toxic effects of fermentation inhibitors.

## 1. Introduction

Bioethanol produced from lignocellulose is an alternative to fossil fuels and other energy sources, and its production from waste materials and energy crops does not pose a risk of rising food and feed prices. Due to the complex structure of lignocellulosic biomass, composed of cellulose, hemicellulose and lignins, pretreatment is a necessary step before enzymatic hydrolysis and fermentation. The aim of pretreatment is to increase the susceptibility of biomass to enzymatic hydrolysis, by reducing the number of crystalline regions in cellulose, and partial degradation of hemicellulose and lignins [1,2]. The pretreatment process can be carried out with the use of various chemical methods (acids, bases, ionic liquids, ozone, organic solvents), physical techniques (grinding, extrusion, use of microwaves, ultrasounds) and physicochemical methods (the use of increased pressure and temperature combined with dilute acids and bases, steam explosion, ammonia fiber explosion, CO_2_ explosion); the pretreatment method should be selected individually for each source of lignocellulosic biomass [3]. During the pretreatment, under the influence of the factors used, toxic by-products can also be formed. These compounds are classified into three groups: weak organic acids (acetic, formic and levulinic acid), furan compounds (5-hydroxymethylfurfural, furfural) and phenolic compounds (syringaldehyde, ferulic acid, vanillin, 4-hydroxybenzoic acid). These compounds are considered inhibitors of yeast metabolism. Furan aldehydes are formed by dehydration of simple sugars. Furfural is formed from xylose which is released from hemicellulose. In turn, 5-hydroxymethylfurfural (5-HMF) is produced from glucose obtained from cellulose and hemicellulose. Acetic acid formation is the result of the release of acetate groups from hemicellulose. Levulinic acid is produced by dehydration of 5-HMF. Formic acid is a product of further transformations of 5-HMF and furfural. Phenolic compounds are released as a result of lignin degradation. Vanillin and vanillic acid are degradation products of guaiacylpropane units of lignin [4,5,6,7,8,9].

The presence of toxic pretreatment by-products in the fermentation medium negatively affected the cellular metabolism of yeast and reduced the fermentation activity (Figure 1). 

The inhibitory effect may be due to the presence of volatile fatty acids (VFAs) such as butyric, propionic and acetic acids, which are in part products yeast metabolism, and/or compounds released during biomass decomposition, such as formic and levulinic acid. Acid stress is defined as the combined biological effect of the interaction of low pH and the presence of weak organic acids in the environment. Weak acids can diffuse undissociated through the cell membrane and dissociate inside the cell, lowering the internal pH value (pHi). This means that at pHo 3.5, a lower concentration of organic acid is required to disintegrate the cells than at pHo 4.4. [10,11]. Aliphatic acids can also enter the yeast cell via the *Fps1p* aquaglyceroporin channel. Weak organic acids dissociate in a neutral intracellular environment, which causes the release of protons and lowers pH of the cytoplasm. Cells react with increased activity of membrane ATPase, which removes protons outside the cell, however, the acetate or formate groups accumulating inside the cell cause damage to the structure and functions of DNA and proteins [12,13,14,15]. Furan aldehydes (furfural and 5-HMF), formed as a result of dehydration of simple sugars, also have a negative effect on yeast metabolism. They reduce the activity of the glycolytic pathway, damage DNA, cell wall and membrane, and inhibit RNA and protein synthesis. In order to reduce the toxicity of furan compounds, yeast cells and bacteria developed a mechanism of aldehyde reduction to the appropriate alcohols. Furfural is reduced to furfuryl alcohol and 5-HMF to 2,5-bishydroxymethylfuran. The stress-stimulated gene *GRE2* encoding 3-methylbutanal reductase and NADPH-dependent methylgloxal reductase is involved in the in-situ detoxification process of furan aldehydes. These biocatalysts enable the conversion of furfural and 5-HMF [5,6,16,17]. It is also believed that NADH-dependent alcohol dehydrogenase participates in the reduction of furan aldehydes [8]. Phenolic lignin degradation products are the most toxic by-products of lignocellulose pretreatment due to their low molecular weight. The mechanism of phenolic compounds’ influence on cellular metabolism has not been fully understood. However, the negative influence of lignin degradation products on the structure and integrity of the yeast cell membrane was confirmed [8]. It should be noted that the by-products of the lignocellulose pretreatment never occur individually, and the synergy of several inhibitors has been observed. The simultaneous presence of acetic acid and furfural increases the toxic effect that inhibits the production of yeast biomass. When, in addition to furfural, vanillin is present in the culture medium, it increases the oxidative stress caused by the former and intensifies the fragmentation of mitochondria [18,19].

Various methods have been developed to reduce the toxic effects of inhibitors of yeast metabolic activity present in the lignocellulose fermentation medium. One of the solutions is to limit the influence of factors promoting an increase in the concentration of inhibitors. This can be done by optimizing the process parameters of the biomass pretreatment. Another solution is to detoxify the medium using chemical, physical or biological methods before the actual fermentation process. One of the current trends is the use of yeast strains with increased tolerance to toxic by-products formed during the pretreatment of lignocellulose [20]. Increased tolerance of cells to toxic stress is achieved by overexpression of genes involved in a specific cellular response. The most commonly used techniques include genetic engineering methods and induced mutagenesis using UV radiation or chemicals. Increased tolerance to pretreatment by-products can even be achieved by overexpression of a single gene. Overexpression of the *ZWF1* gene encoding glucose-6-phosphate dehydrogenase led to increased tolerance to high furfural concentrations, while overexpression of the *ADH6* gene encoding NADPH-dependent alcohol dehydrogenase improved the tolerance to high levels of 5-HMF [21,22,23]. Increased resistance to inhibitors of fermentation processes can also be obtained by manipulating multiple genes involved in the cell’s response to toxic stress using the global transcription engineering technique (gTME) [24,25]. An alternative to genetic engineering techniques are adaptation processes carried out under conditions of toxic stress, providing cells with increased tolerance to fermentation inhibitors. Adaptation is a very useful strategy in constructing a population of cells with an altered cellular metabolome (higher content of individual intracellular metabolites), guaranteeing increased resistance to stress factors. It was demonstrated that the presence of an increased concentration of furfural and 5-HMF in the culture medium boosted the expression of genes involved in their metabolism. Yeast populations with increased tolerance to inhibitors can grow on media with an elevated concentration of lignocellulose hydrolysates and enter the early fermentation phase faster; the total duration of the process was shorter [26,27,28]. However, the use of adaptive techniques to obtain a population of yeast with increased tolerance to stressors in the fermentation medium (inhibitors generated during the pretreatment of lignocellulose) requires a more complete understanding of the specific response of yeast cells, manifested by the production of intra- and extracellular metabolites such as ergosterol, trehalose, glycerol, and specific proteins [29].

There is relatively little experimental data showing the effect of inhibitors formed during the pretreatment of lignocellulosic biomass on the production of intra- and extracellular metabolites by yeast cells. Similarly, little is known of the production of heat shock proteins (HSPs) as a specific cellular response of *S. cerevisiae* to exposure to toxic inhibitors of fermentation processes. A better understanding of these phenomena was the aim of the presented study. The object of the study on the response of yeast cells was the popular *S. cerevisiae* Ethanol Red strain. The analysis covered the production of intracellular stress indicators (trehalose, ergosterol, total proteins) and the extracellular concentration of acetic acid and glycerol as a response to the presence of toxic stressors such as levulinic acid, furfural, 5-HMF, ferulic acid, syringaldehyde and vanillin. The experiment was carried out both during aerobic cultivation and under alcoholic fermentation conditions. One of the aspects of the novelty is an attempt to determine the participation of HSPs (Hsp31p and Hsp60) in the reaction of yeast cells to the presence of toxic substances resulting from the pretreatment of lignocellulose. The experiments under both aerobic and anaerobic conditions were carried out using only one toxic stressor at three different concentrations or with the simultaneous use of several stressors to assess the intensity of the synergistic interaction. The results showing the ability of yeast to detoxify lignocellulosic culture media in situ during 72 h cultivation under various conditions are also important not only for scientific but also application reasons.

## 2. Results

### 2.1. Assessment of the Capacity of S. cerevisiae Strain Ethanol Red to Metabolize Fermentation Inhibitors (By-Products of Pretreatment of Lignocellulosic Biomass)

Analysis were performed on model media with the toxic by-products of lignocellulose pretreatment added at three concentrations as shown in Table 1. Yeast cells were grown under aerobic or alcoholic fermentation conditions for 72 h. The concentrations of the by-products used were selected on the basis of previous studies. They corresponded to the actual levels observed during the pretreatment of various sources of lignocellulosic biomass [30,31]. Under aerobic conditions, the concentration of glucose (carbon source) in the medium was 20 g/L and no formation of ethanol was observed. Alcoholic fermentation was induced using the Crabtree effect by increasing the glucose concentration in the medium to 200 g/L. Under such conditions, the alcoholic fermentation led to a complete bioconversion of glucose to ethanol. The substrates were contaminated with inhibitors selectively, and using combinations thereof as shown in Table 1. Target concentrations of these compounds in the culture media were monitored by HPLC before adding the yeast and then after 72 h of the culture. Percent concentration reduction of the individual compounds was calculated based on the concentration difference.

The *S. cerevisiae* strain Ethanol Red, regardless of the culture conditions (aerobic or anaerobic) and the concentration of inhibitors, was able to completely metabolize 5-HMF, furfural, syringaldehyde and vanillin, if the substances were added individually (Table 1). The utilization of levulinic acid by the yeast strain used depended on the culture conditions and the concentration of this acid. Under aerobic conditions, a higher level of levulinic acid concentration reduction was observed. It was 42.9% at the lowest initial concentration (408.1 mg/L) and decreased to 22.3% at the highest concentration used (1606.4 mg/L) (Table 1). Under the conditions of alcoholic fermentation, the reduction of levulinic acid concentration in the medium was smaller and amounted to a maximum of 16% for the lowest initial concentration of this acid. A similar tendency was found for ferulic acid. Higher concentration reduction of this acid (100%) was observed for aerobic conditions with the lowest degree of contamination (83.4 mg/L). As the concentration of the initial ferulic acid increased, its percentage reduction decreased: to 89.8% for the initial concentration of 176.0 mg/L and to only 36% for the initial concentration of 376.3 mg/L. Under the conditions of alcoholic fermentation, the yeast’s ability to metabolize this compound was smaller and amounted to a maximum of about 55.3% for the lowest initial contamination level of 108.0 mg/L (Table 1). The analysis of interactions caused by simultaneous contamination of culture media with several inhibitors showed a lower degree of reduction of the concentrations of these substances compared to media containing only one inhibitor. The simultaneous presence of levulinic acid, 5-HMF and furfural in the medium, regardless of the culture conditions, reduced the ability to metabolize levulinic acid to only 9.1%. Contamination of the medium with 5-HMF and furfural slightly decreased the ability to utilize 5-HMF under aerobic conditions, by ca. 5%. The coexistence of the analyzed lignin degradation products, regardless of the cultivation conditions, did not reduce the ability of yeasts to metabolize them (Table 1). Simultaneous contamination of the culture medium with all the analyzed by-products of lignocellulose pretreatment under aerobic conditions decreased the ability to utilize levulinic acid, 5-HMF and ferulic acid. Under the conditions of alcoholic fermentation only a decrease in the ability to metabolize ferulic acid was observed (Table 1). Our study confirmed that the industrial yeast strain used during fermentation exhibited a high tolerance to toxic lignocellulose pretreatment by-products, and was able to metabolize these compounds to less toxic substances.

### 2.2. Effect of By-Products of Lignocellulose Pretreatment on the Growth of Yeast Biomass and the Concentration of Intracellular Metabolites under Various Culture Conditions

Under aerobic conditions, the presence of levulinic acid and lignin degradation products (ferulic acid, syringaldehyde, vanillin) in the culture medium had no statistically significant effect on the biomass. As compared to control, a negative influence of the culture medium contamination with furan aldehydes, mainly furfural, on the yeast biomass growth rate, was observed (Figure 2A).

In contrast to the aerobic conditions, yeast biomass growth during alcoholic fermentation was mainly influenced by phenolic compounds (lignin degradation products). This effect was probably related to the greater permeability of the cell membrane caused by the increasing concentration of ethanol (Figure 2A). The simultaneous presence of furan aldehydes (5-HMF and furfural) under aerobic conditions significantly reduced the yeast biomass growth (by ca. 6 mg/mL). This effect was mitigated by levulinic acid (biomass concentration was lower by about 1.2 mg/mL) (Figure 2B). Under the conditions of alcoholic fermentation, the combined presence of furan aldehydes or phenolic compounds did not have a statistically significant influence on the growth of biomass. On the other hand, the presence of all analyzed inhibitors in the medium, regardless of the type of culture, caused a slight decrease in biomass concentration after 72 h of culture, by ca. 1–2 mg/mL (Figure 2B).

The presence of one lignocellulose pretreatment by-product in the culture medium caused changes in the concentration of intracellular proteins. Under aerobic conditions, the highest concentration of intracellular proteins was observed when the medium was contaminated with ferulic acid (O^+^Fer^−1^, O^+^Fer^0^), 5-HMF (O^+^5-HMF^−1^) and furfural (O^+^Fur^+1^). On the other hand, increased concentration of levulinic acid (O^+^LA^0^, O^+^LA^+1^) in the culture medium inhibited the biosynthesis of intracellular proteins (Figure 3A).

During the alcoholic fermentation, a significant increase in the amount of intracellular proteins (to the level of ca. 50 µg/mg of biomass) was found only for the increased concentration of ferulic acid (O^−^Fer^+1^). Other analyzed inhibitors, added individually to the fermentation medium, had little effect on changes in the amount of intracellular proteins produced (Figure 3A). The simultaneous presence of several inhibitors induced an overproduction of intracellular proteins, regardless of the type of culture. The synergism of the toxic interaction was most clearly marked when 5-HMF and furfural were used simultaneously under aerobic conditions. The concentration of intracellular proteins increased then to about 80 µg/mg of biomass. In other experimental variants, the simultaneous use of a greater number of by-products resulted in the production of intracellular proteins in the range from 38 to 47 µg/mg of biomass, which was almost twice as high as in the control variants (Figure 3B).

The biosynthesis of ergosterol, which is an indicator of cellular stress, was also strongly dependent on the culture conditions. The concentration of ergosterol in yeast cells was higher under aerobic conditions than during alcoholic fermentation (Figure 4A). Contamination of the medium with levulinic acid, furfural or ferulic acid limited the biosynthesis of this metabolite under aerobic conditions. The lowest concentration of ergosterol was observed when furfural was the contaminant, regardless of its concentration. Syringaldehyde and vanillin had no statistically significant effect on the concentration of ergosterol in the cell, while 5-HMF caused an overproduction of this intracellular metabolite when oxygen was available (Figure 4A).

In turn, during the alcoholic fermentation, the highest concentration of ergosterol in the cell was observed when ferulic acid was present in the medium. The lowest concentration of this metabolite in cells was found at the highest concentration of 5-HMF (O^+^5-HMF^+1^) and syringaldehyde in the culture medium (Figure 4A). There was no synergistic effect of sugar dehydration products (levulinic acid, 5-HMF, furfural) or lignin degradation products (ferulic acid, syringaldehyde, vanillin) on the concentration of ergosterol in the cell, regardless of the culture conditions (Figure 4B). On the other hand, when all the analyzed inhibitors were present in the medium, ergosterol biosynthesis under aerobic conditions was reduced by nearly half, i.e., to the level of ca. 35 µg/mg of biomass, compared to the control variant (Figure 4B).

Similar relationships as in the case of ergosterol were observed for changes in trehalose concentration in the cell. In general, higher intracellular concentrations of this disaccharide were found under aerobic conditions. The presence of levulinic acid did not affect the amount of trehalose in the cell, regardless of the culture conditions (Figure 5A). Under aerobic conditions, the lowest concentration of trehalose was found when furfural was present in the culture medium, which was associated with the limitation in the biosynthesis of this disaccharide or an increased activity of acid trehalase. Under aerobic conditions, lower concentrations of 5-HMF (O^+^5-HMF^−1^, O^+^5-HMF^0^) decreased intracellular trehalose concentration to the level of 66 µg/mg of biomass.

At the highest concentration of 5-HMF, the content of trehalose increased to the level of ca. 120 µg/mg of biomass, i.e., it was ca. 20% higher than in the control variant (Figure 5A). The highest concentration of ferulic acid (O^+^Fer^+1^) in the medium significantly reduced the concentration of trehalose in the cell under aerobic conditions. During alcoholic fermentation, the lowest concentration of trehalose (ca. 6.5 µg/mg of biomass) was observed at the highest concentration of 5-HMF (O^+^5-HMF^+1^) and ferulic acid (O^+^Fer^+1^) (Figure 5A). The highest concentration of this disaccharide (ca. 110 µg/mg of biomass) was found for the highest contamination of the substrate with furfural (O^+^Fur^+1^). The metabolic reaction of yeast cells to the simultaneous presence of several lignin degradation products under aerobic conditions was the overproduction of trehalose to the level of ca. 140 µg/mg of biomass. In turn, under the conditions of alcoholic fermentation, a reduced concentration of trehalose to about 20 µg/mg of biomass was observed (Figure 5B). The synergistic toxic effect of sugar dehydration products was evident especially during fermentation. It resulted in a higher concentration of trehalose by more than 40 µg/mg of biomass, compared to the control variant. In turn, in the conditions of aerobic culture, the concentration of this disaccharide was lower by ca. 60 µg/mg of biomass, compared to the control variant (Figure 5B).

### 2.3. Effect of By-Products of Lignocellulose Pretreatment on the Production of Extracellular Metabolites (Acetic Acid and Glycerol) under Aerobic Conditions and during Fermentation

Glycerol and acetic acid are metabolites produced during alcoholic fermentation. Glycerol is formed by the reduction of dihydroxyacetone phosphate to glycerol-3-phosphate, which is then dephosphorylated. The main role of glycerol is to balance the intracellular redox potential by converting excess NADH to NAD^+^ [32]. Glycerol also participates in the yeast cell’s response to osmotic stress, reducing the difference between intra- and extracellular osmotic pressure [33]. Acetic acid can be formed in acetyl-CoA hydrolysis. At an earlier stage, pyruvate dehydrogenase is involved in the formation of acetyl-CoA by oxidative decarboxylation of pyruvic acid. This reaction takes place in the mitochondrial matrix, but under anaerobic conditions is limited. Due to the anaerobic conditions accompanying fermentation, acetic acid is formed mainly by oxidation of acetaldehyde with aldehyde dehydrogenase and NADP^+^ as a cofactor. The resulting NADPH participates in lipid biosynthesis [34]. The above explains why these extracellular metabolites were found only in those experimental variants where ethanol biosynthesis was observed. The presence of 5-HMF and furfural in the culture media stimulated the alcohol fermentation process, which was probably due to the increased activity of alcohol dehydrogenase involved in the metabolism of furan aldehydes [22]. Under aerobic conditions, the production of extracellular glycerol was found only in the media with 5-HMF and those with furfural at its highest concentration (O^+^Fur^+1^) (Figure 6A). During fermentation, lower glycerol production was observed compared to the control variant, with the exception of the medium with the highest concentration of 5-HMF (O^+^5-HMF^+1^). The lowest concentration was found in the medium with furfural at the highest concentration (O^+^Fur^+1^), and in those with ferulic acid at the two lowest concentrations (O^+^Fer^−1^, O^+^Fer^0^) (Figure 6A).

Similar trends in the concentration of glycerol in the culture medium were also observed for the simultaneous contamination with several by-products of lignocellulose pretreatment (Figure 6B).

Acetic acid in the medium under aerobic conditions was found only in the case of furfural contamination (Figure 7A). During fermentation, this acid was present in all media, but only contamination with syringaldehyde (in different concentrations) doubled its production compared to the control medium. Contamination with the other analyzed by-products had little effect on the acetic acid concentration (Figure 7A). Under the conditions of alcoholic fermentation, contamination with furan aldehydes or phenolic compounds reduced the formation of this acid by ca. 0.3 g/L compared to the control variant (Figure 7B).

### 2.4. Production of Hsp31p and Hsp60 as a Metabolic Response of Yeast Cells to Stress Caused by the Presence of Lignocellulose Pretreatment By-Products

In response to toxic stress, yeast cells can synthesize heat shock proteins (HSPs), which are involved in forming and stabilizing other cellular proteins. In this study, the relationship between the contamination of the culture medium by by-products of lignocellulose pretreatment and the biosynthesis of Hsp31p and Hsp60 was investigated. Hsp31p is believed to be involved in the response to oxidative stress, and Hsp60 supports the folding of cellular proteins [35,36].

Immunodetection analysis showed the presence of Hsp31p and Hsp60 in yeast cells in all control and experimental variants, regardless of whether the culture conditions were aerobic or anaerobic (during alcoholic fermentation) (Figure 8). The presence of HSPs in yeast cells grown on media not contaminated with the by-products of lignocellulose pretreatment indicates the physiological role of these macromolecules in *S. cerevisiae* strain Ethanol Red. The western blot method used, combined with immunodetection, allowed only qualitative analysis, however, based on the intensity of the bands obtained, an upward or downward trend in the concentration of the analyzed proteins can be determined. Hsp31p was synthesized in all experimental variants, but under aerobic conditions its concentration was higher, which resulted from the participation of this protein in processes related to oxidative stress (Figure 8). The analysis of the presence of Hsp60 showed an increased contribution of this polypeptide in the yeast response to stress caused by the presence of levulinic acid, 5-HMF, furfural, ferulic acid, syringaldehyde and vanillin, regardless of the culture conditions (Figure 8). An increase in the production of these proteins was observed with increasing concentrations of 5-HMF, furfural, ferulic acid, syringaldehyde and vanillin under aerobic conditions, and ferulic acid during alcoholic fermentation. In yeast cultures on media contaminated simultaneously with all lignin degradation products (ferulic acid, syringaldehyde and vanillin), increased synthesis of this group of proteins was seen for both aerobic and anaerobic cultures (Figure 8).

## 3. Discussion

In this study, we analyzed the specific cellular response of the *S. cerevisiae* strain Ethanol Red to the toxic stress caused by the presence of inhibitors in the culture medium, which are formed as by-products during the pretreatment of lignocellulosic biomass. The analysis included the effects caused by the presence of inhibitors in the model media in the concentrations actually observed in the hydrolysates, which were produced as a result of thermochemical treatment of various lignocellulosic raw materials (Table 1). The observations were carried out on yeast grown under aerobic conditions and during alcoholic fermentation (the presence of ethanol was confirmed by the analysis of the medium composition after 72 h of the culture, unpublished results). An industrial yeast strain with increased tolerance to osmotic, ethanol and thermal stress used in the production of first-generation ethanol was selected for the study. Experiments also confirmed the tolerance of this yeast strain to an increased concentration of pretreatment by-products. This was evidenced by the low influence of inhibitors on the yeast biomass growth rate under aerobic conditions and the results of glucose conversion into ethanol during fermentation. Regardless of the culture conditions, the Ethanol Red strain showed the ability to completely utilize 5-HMF, furfural, syringaldehyde and vanillin and partially metabolize levulinic and ferulic acids. The ability of yeast to metabolize furan aldehydes to less toxic derivatives is due to the activity of aldehyde reductases, including alcohol dehydrogenase. The high activity of this enzyme is a characteristic feature of industrial yeast strains used in alcoholic fermentation processes [37]. Also the *S. cerevisiae* NRRL Y-12632 strain was able to metabolize 5-HMF under aerobic conditions. This aldehyde, initially present in the model medium at a concentration of ca. 28 mmol, was completely eliminated after 48 h. However, an increased activity of aldehyde reductases converting 5-HMF to 2,5-bis-hydroxymethylfuran was observed only after 35 h of the culture [16]. Also the mutants Y62-C11 and Y62-G6 of the strain *S. cerevisiae* NRRL Y-12632 showed the ability to metabolize 5-HMF, but the complete utilization of this aldehyde from ca. 26 mmol took ca. 90 h with galactose as the source of carbon [17]. It is not only the *S. cerevisiae* that is able to metabolize 5-HMF. Another yeast species used in the production of bioethanol from lignocellulose is *Scheffersomyces stipitis* (*Pichia stipitis*) consuming xylose as a carbon source, which also reduce the concentration of 5-HMF in the culture medium. The *S. stipitis* yeast, like the Ethanol Red strain used in this work, demonstrated the ability to reduce the concentration of 5-HMF, but in the higher concentration range, from 5 to 10 g/L, during fermentation process lasting up to 100 h. The similarity also applies to its limited ability to metabolize levulinic acid. Under the conditions of the alcoholic fermentation of pretreated *Gelidium amansii* seaweed, the concentration of levulinic acid remained constant for up to 155 h of the process [5]. Furfural can be metabolized not only by yeast, but also by bacteria with high aldehyde reductase activity, used in the production of biobutanol during anaerobic fermentation. *Clostridium acetobutylicum* ATCC 824 bacteria were able to completely metabolize furfural to furfuryl alcohol during fermentation, the main products of which are acetone, butanol and ethanol [18]. The high toxicity of vanillin, which stimulated the accumulation of ROS (reactive oxygen species) in yeast cells, contributed to mitochondrial fragmentation and translation inhibition by blocking ribosome assembly, which significantly reduced the intensity of yeast metabolism [19,38]. The activity of alcohol dehydrogenase mitigates the negative influence of vanillin on yeast cells. This enzyme is encoded by the *ADH6* gene. It has low substrate specificity, including also the conversion of vanillin to vanillin alcohol [39]. High activity of dehydrogenases participating in the processes of acetaldehyde reduction to ethyl alcohol is a feature of industrial distillery yeast, including the Ethanol Red strain. Probably due to these properties, the tested strain fully utilized vanillin in a wide range of concentrations, from 20 to 80 mg/L, regardless of the culture conditions. Other *S. cerevisiae* strains overexpressing the *ADH6* gene also showed the ability to completely reduce the concentration of vanillin from as much as 1 g/L during 40 h of culture [40].

Earlier publications on the effect of toxic by-products generated during the pretreatment of lignocellulosic biomass on yeast cellular response focused mainly on transcriptomic analysis, which allowed to determine the degree of gene expression in response to environmental stress factors [41,42]. To the knowledge of the authors of this study, no comprehensive analysis of the influence of these toxic compounds (inhibitors) on the accumulation of intracellular metabolites, i.e., trehalose or ergosterol have been performed so far. Our studies showed that vanillin at the analyzed concentrations (from 20 to 80 mg/L of medium) had no effect on the concentration of ergosterol in yeast cells. Ergosterol, which is a component of the cell membrane, is responsible for its fluidity. It is also believed to be responsible for yeast tolerance to elevated vanillin levels. Studies on the *S. cerevisiae* NBRC 1950 strain, characterized by increased tolerance to the presence of vanillin, supported this view. The concentration of intracellular ergosterol in the cells of this microorganism was almost twice as high as in the strains not resistant to the presence of vanillin in the culture medium. However, such concentration of ergosterol was observed for a very high concentration of vanillin, 2.28 g/L during strain selection procedure [39]. At the increased concentration of 5-HMF in the conditions of aerobic culture, an overproduction of ergosterol was observed. Increased production of ergosterol, one of the main components of the yeast cell membrane, is a response to oxidative stress and aims to maintain the integrity of the cell membrane under adverse environmental conditions [43]. In our study, increased intracellular concentration of trehalose in yeast cells was found in the media containing 5-HMF under aerobic conditions and furfural under alcoholic fermentation conditions. Undoubtedly, this is related to one of the defense mechanisms aimed at preventing ROS-induced damage. Accumulation of trehalose protects the integrity of the cell membrane in toxic stress caused by furan aldehydes [44].

In response to the toxic stress caused by the presence of by-products from the pretreatment of lignocellulosic biomass, yeast cells can also produce chaperones. In our studies, we observed a higher concentration of Hsp31p in yeast grown under aerobic conditions compared to cultures under alcoholic fermentation conditions, which confirms the participation of these proteins in the response to oxidative stress. Moreover, an overproduction of this protein in yeast was found during the aerobic culture in the presence of furfural, which is believed to be a compound that generates oxidative stress. The presented results are consistent with the findings of other authors [35], reporting the role of the Hsp31p in the *S. cerevisiae* response to stress induced by reactive oxygen species. In our research, we focused mainly on the presence of the Hsp31p and Hsp60 involved in the metabolic reaction of *S. cerevisiae* yeast to oxidative stress, caused by compounds such as furan aldehydes, formed during the processing of lignocellulosic biomass. The role of Hsp60 under oxidative stress is to prevent denaturation of Fe/S proteins participating in in the mitochondrial electron transport chain. The overproduction of mitochondrial Hsp60 helps to reduce protein denaturation and their aggregation, which has a positive effect on the viability of yeast cells in a wide temperature range [45]. This knowledge may be helpful in the development of a genetically modified strain of *S. cerevisiae* with increased tolerance to toxic stressors. As shown in the present study, the reaction of yeasts to the presence of oxidative stress-generating furan aldehydes in the culture medium is the overproduction of Hsp60. These proteins are involved in the folding of intracellular proteins [46]. To better understand the role of HSP in the response to toxic stress induced by inhibitors of lignocellulose pretreatment, we plan to conduct further studies involving Hsp40, Hsp70, Hsp90 and Hsp104 involved in the response to thermal and ethanol stress [47]. 

## 4. Materials and Methods

### 4.1. Yeast

The distillery yeast *Saccharomyces cerevisiae* strain Ethanol Red (Lesaffre Advanced Fermentations, Marcq-en-Barœul, France) was used in the experiments in the form of active dried yeast preparation. The microorganisms were applied in the form of yeast milk prepared by suspending 1 g of the preparation in 10 mL of sterile 0.9% *v*/*v* NaCl at 30 °C (according to the manufacturer’s instructions). The cell count was 1.25 ± 0.12 × 10^9^ CFU/mL with viability of 94.3 ± 0.5%. The yeast dose was 200 µL of yeast milk per 100 mL of culture medium.

### 4.2. Materials

Analytical reagents, high purity media components and HPLC grade solvents used in the study were supplied by Merck^®^ (Darmstadt, Germany). The standards used in the chromatographic analyzes were HPLC grade, supplied by Sigma-Aldrich^®^ (St. Louis, MO, USA). All by-products of lignocellulose pretreatment, i.e., levulinic acid, 5-HMF, furfural, ferulic acid, syringaldehyde and vanillin, were added to the culture media as ethanol solutions.

### 4.3. Research Plan

The research was carried out on the cultures of the *S. cerevisiae* on model media contaminated with toxic stressors in three concentrations (Table 1). The yeast cultures were grown for 72 h under aerobic or alcoholic fermentation conditions. The aerobic cultures were grown in medium containing glucose as a carbon source at a concentration of 20 g/L in a water bath with shaking (80 rpm) at 28 °C in conical flasks with septa intensifying the culture aeration. The alcoholic fermentation process was forced by the use of an elevated glucose concentration of 200 g/L (Crabtree effect). The cultures were grown in conical flasks placed in a shaking water bath (80 rpm) at 28 °C. All cultures were carried out in triplicate. All experimental variants are presented in Table 1.

### 4.4. Culture Media

Synthetic culture media were used to analyze the effect of environmental stress on the metabolism of *S. cerevisiae*. The composition of the synthetic culture media used during aerobic cultivation was as follows: glucose 20.0 g/L, peptone 3.5 g/L, yeast extract 3.0 g/L, KH_2_PO_4_ 2.0 g/L, (NH_4_)_2_SO_4_ 1.0 g/L, MgSO_4_ × 7H_2_O 1.0 g/L, chloramphenicol 0.1 g/L [48]. Cultures in alcoholic fermentation conditions were carried out on the same media, but with the glucose concentration increased to 200 g/L. Appropriate amounts of lignocellulose pretreatment by-products were added to the media before inoculation with yeast milk.

### 4.5. Analytical Methods

Intracellular concentration of trehalose, ergosterol, protein, biomass and the extracellular concentration of glycerol and acetic acid were analyzed after 72 h of the culture. The concentration of by-products of lignocellulose pretreatment in the culture medium was determined before starting the culture and after 72 h of incubation at 28 °C.

#### 4.5.1. Determination of Yeast Biomass Concentration

The concentration of yeast biomass was determined after 72 h of the culture by measurement of absorbance at 600 nm [49]. The analyzed biomass sample was centrifuged at 7000 g for 10 min at 20 °C, rinsed with sterile 0.9% *w/v* NaCl and centrifuged. The obtained pellet was suspended in 0.9% *w/v* NaCl and the biomass concentration was measured. Biomass concentration was measured based on calibration curve prepared from the dried yeast suspension.

#### 4.5.2. Determination of Intracellular Concentration of Trehalase

The intracellular concentration of trehalose was determined by high performance liquid chromatography (HPLC) after cell lysis [50]. The analysis was started with centrifugation of yeast biomass at 7000 g for 5 min at 20 °C. Pellet was then rinsed with sterile 0.9% *w/v* NaCl and centrifuged again (conditions as above). After supernatant removal, the cells were lysed by microwave irradiation (cells were incubated 3 times under microwave irradiation at 700 W for 1 min with a 30 s interval). The lysed cells were shaken for 2 h at 200 rpm with a 5 mM H_2_SO_4_ solution which was used as the mobile phase for the determination of the sugar concentration using the HPLC kit. Before analysis, the obtained solution was filtered through a 0.45 µm pore size membrane filter. The analysis of the trehalose concentration was performed using a model 1260 chromatograph (Agilent Technologies^®^, Palo Alto, CA, USA), equipped with a refractometric detector. The chromatographic separation was performed using a Hi-Plex H column (Agilent Technologies^®^, Palo Alto, CA, USA) equipped with a dedicated guard column, with a mobile phase flow of 0.6 mL/min. at 60 °C. Quantitative analysis was carried out based on the ESTD calibration.

#### 4.5.3. Determination of the Intracellular Concentration of Ergosterol

After chemical lysis of yeast, the concentration of intracellular ergosterol was determined using the spectrophotometric method [47]. Yeast cells were centrifuged at 7000 g for 5 min at 20 °C. Then the supernatant was removed, the cells were rinsed with 0.9% *w/v* NaCl and centrifuged (conditions as above). The cell suspension in a solution of 25% *w/v* KOH in ethanol was incubated for 4 h in a water bath at 90 °C. The solution, after cooling to 20 °C, was diluted with distilled water (1 mL of distilled water was added to 2 mL of the solution), and 5 mL of n-heptane was added. The solution was then shaken for 3 min. The organic phase was taken for analysis. The concentration of ergosterol was determined by spectrophotometry at 283 nm against the standard curve.

#### 4.5.4. Determination of Intracellular Protein Concentration

Intracellular protein concentration was measured by Bradford method after cell lysis [51]. The cell lysis procedure was started by centrifuging the cells at 7000 g for 15 min at 20 °C. Pellet was suspended in distilled water at 4 °C and aliquoted to centrifuge tubes. The solution was centrifuged (7000 g for 15 min at 20 °C) and the pellet was stored at −80 °C. Frozen cells were suspended in 0.1 M NaOH and incubated at 20 °C for 5 min. The resulting solution was centrifuged (7000 g for 15 min at 20 °C). After removal of the supernatant, cells were resuspended in lysis buffer (4% SDS, 20% glycerol, 120 mM Tris-HCl pH 6.8, 10% 2-mercaptoethanol added before lysis started) and incubated at 80 °C for 8 min. The solution was centrifuged (7000 g for 5 min at 20 °C). The obtained supernatant was used for protein determination and HSP analysis by western blot [49].

#### 4.5.5. Determination of HSP Proteins by Western Blot

Analyzes were started with SDS-PAGE electrophoresis and protein transfer to PVDF membrane [52]. The wells were loaded with 20 µg of the protein. The transfer of proteins to the PVDF membrane was performed using the BioRad^®^ (Hercules, CA, USA) semi-dry transfer system with a kit containing the membrane, blotting paper and transfer buffer. The membrane was incubated for 60 min in blocking buffer (TBST) containing 3% bovine albumin, and then soaked in TBST for 10 min. The next step was an overnight incubation at 4 °C with the primary antibody. Primary antibodies were dedicated to *S. cerevisiae* yeast. Dilutions were selected according to the manufacturer’s protocol (Anti-Hsp31p-Host: goat, NSJ Bioreagents, San Diego, CA, USA; Hsp60 Antibody—Host: mouse, OriGene, Rockville, Maryland, USA). After incubation, the membrane was rinsed with TBST buffer (× 5) and incubated for 60 min at 20 °C with secondary antibodies (Peroxidase-Affini Pure Rabbit Anti-Goat IgG, Jackson ImmunoResearch, Cambridgeshire, United Kingdom; Goat Anti-Mouse IgG (H + L)—HRP Conjugate, BioRad^®^, Hercules, CA, USA). The peroxidase was then visualized using a chemiluminescence detection solution (Clarity Western ECL Substrate, BioRad^®^) and the ChemiDoc MT Imager visualization system from BioRad^®^ (Hercules, CA, USA).

#### 4.5.6. Determination of Extracellular Concentrations of Acetic Acid and Glycerol

The extracellular concentrations of acetic acid and glycerol were measured by HPLC. After the culture, the media were centrifuged to remove yeast cells (7000 g for 10 min at 20 °C), diluted with 5 mM H_2_SO_4_ and filtered through a 0.45 µm pore size membrane filter. The concentrations of acetic acid and glycerol were measured using an Agilent Technologies^®^ chromatograph, model 1260, equipped with a refractometric detector. The chromatographic separation was carried out using a Hi-Plex H column (Agilent Technologies^®^) equipped with a dedicated guard column, with a mobile phase flow of 0.6 mL/min. at 60 °C. Quantitative analysis was performed based on the ESTD calibration.

#### 4.5.7. Measuring the Concentration of Lignocellulose Pretreatment By-Products in Culture Media

Levulinic acid concentration was analyzed according to the procedure described in Section 4.5.6. Concentrations of other lignocellulose pretreatment by-products were measured using an Agilent Technologies^®^ (Palo Alto, CA, USA) model 1260 liquid chromatograph equipped with a diode array detector. Separation was done on a ZORBAX Eclipse Plus C_18_ column (4.6 × 100 mm, 3.5 µm; Agilent Technologies^®^, Palo Alto, CA, USA); mobile phase 0.3% acetic acid (70%) and methanol (30%), flow rate 0.5 mL/min., temperature 30 °C. Furfural, 5-HMF, and vanillin were detected at 280 nm. Syringaldehyde and ferulic acid were detected at 320 nm [53]. The quantitative analysis was performed using the ESTD method.

### 4.6. Statistical Methods

All laboratory analyzes were performed in triplicate. Statistical analysis (analysis of variance, determination of SD) was carried out using the TIBCO Software Inc. Statistica ver. 13 (Palo Alto, CA, USA). An ANOVA test and a HSD Tukey’s test were applied at the significance level of α < 0.05.

## 5. Conclusions

To sum up, the presence of by-products of lignocellulose pretreatment in the culture medium, such as levulinic acid, 5-HMF, furfural, ferulic acid, syringaldehyde and vanillin, caused changes in the metabolism of the *S. cerevisiae* strain Ethanol Red. The cellular response depended largely on the culture conditions (aerobic culture or alcoholic fermentation). This study confirmed the possibility of in situ detoxification of culture media with the analyzed yeast strain. Due to the high activity of reductases, efficient reduction of furan aldehydes, syringaldehyde and vanillin was observed, regardless of the applied conditions. In the analyzed concentration range, the highest reduction was found for furan aldehydes and ferulic acid. The experiments also demonstrated the involvement of heat shock proteins in the cellular response induced by fermentation inhibitors. A more complete understanding of the reaction of yeast cells used in alcoholic fermentation to the presence of toxic by-products of lignocellulose pretreatment may facilitate the selection of adaptive culture conditions aimed at obtaining a yeast population with increased tolerance to these toxins.

## Figures and Tables

**Figure 1 molecules-26-00806-f001:**
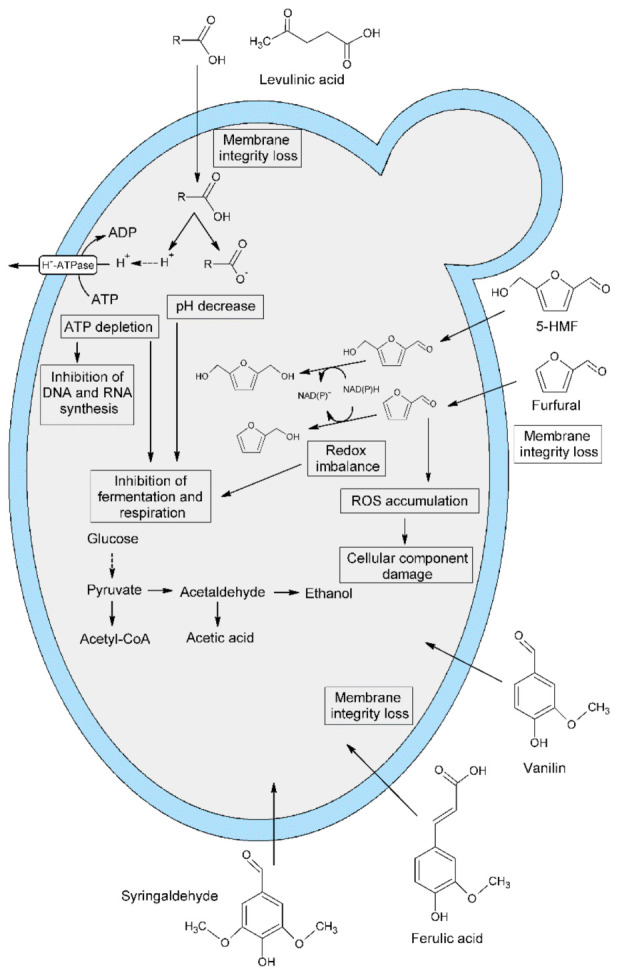
Effect of lignocellulose pretreatment by-products on yeast cell metabolism.

**Figure 2 molecules-26-00806-f002:**
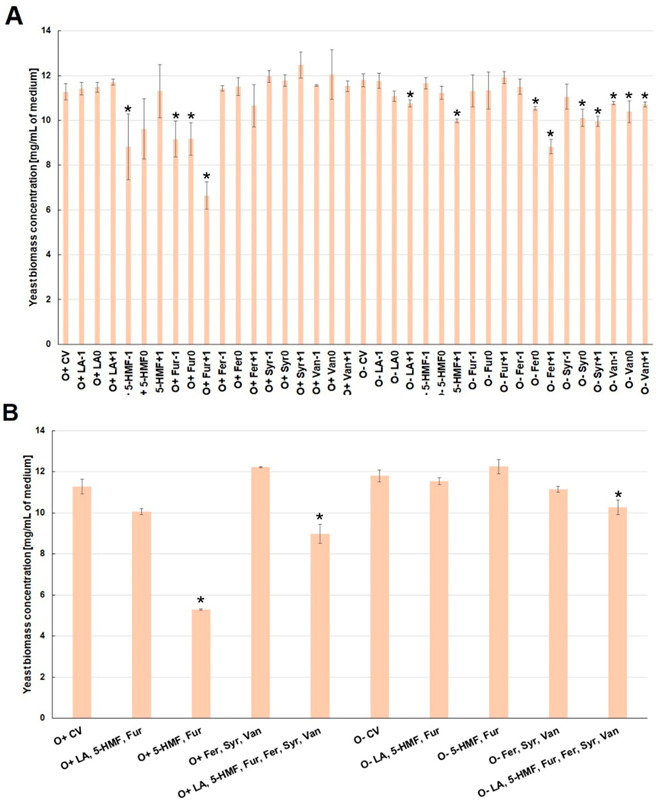
Effect of lignocellulose pretreatment by-products (**A**,**B**) on the yeast biomass production during the culture (*—statistically different from the corresponding control variant α < 0.05).

**Figure 3 molecules-26-00806-f003:**
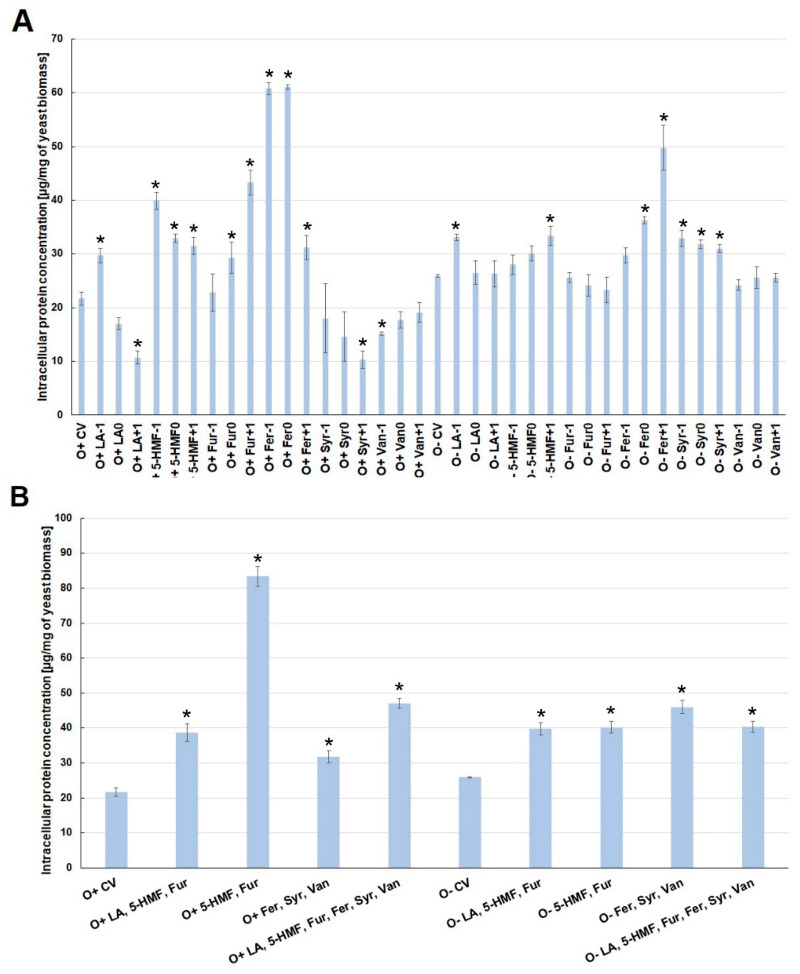
Effect of lignocellulose pretreatment by-products (**A**,**B**) on the production of intracellular proteins by yeast (*—statistically different from the corresponding control variant α < 0.05).

**Figure 4 molecules-26-00806-f004:**
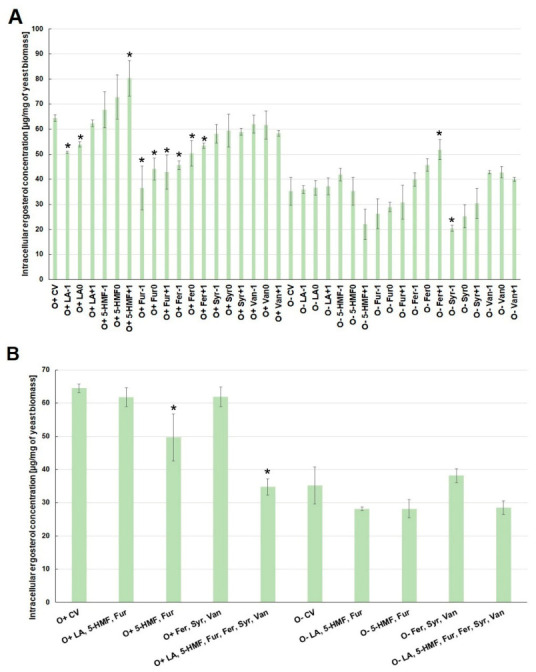
Effect of lignocellulose pretreatment by-products (**A**,**B**) on the production of intracellular ergosterol by yeast (*—statistically different from the corresponding control variant α < 0.05).

**Figure 5 molecules-26-00806-f005:**
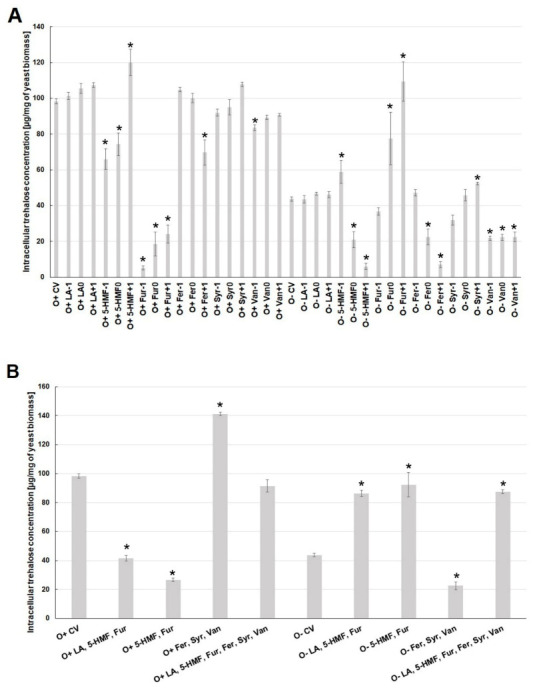
Effect of lignocellulose pretreatment by-products (**A**,**B**) on the production of intracellular trehalose by yeast (*—statistically different from the corresponding control variant α < 0.05).

**Figure 6 molecules-26-00806-f006:**
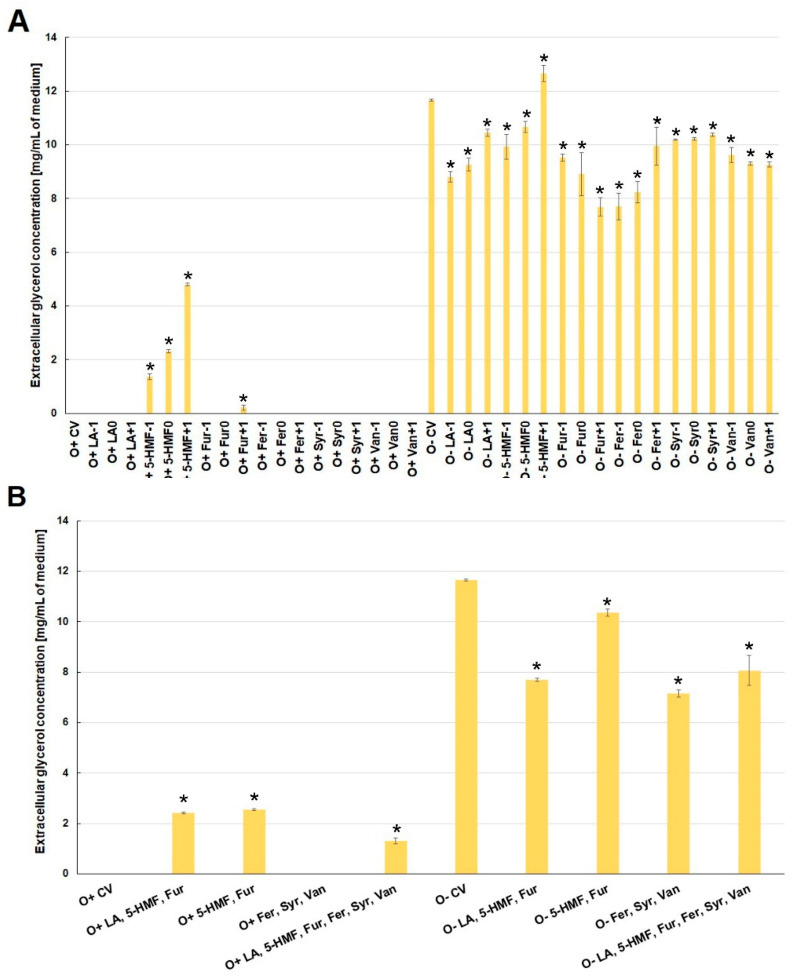
Effect of lignocellulose pretreatment by-products (**A**,**B**) on the production of extracellular glycerol by yeast (*—statistically different from the corresponding control variant α < 0.05).

**Figure 7 molecules-26-00806-f007:**
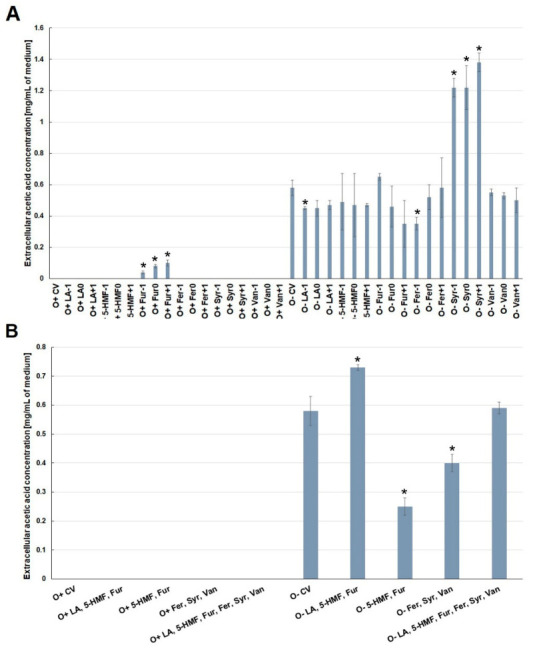
Effect of lignocellulose pretreatment by-products (**A**,**B**) on the production of extracellular acetic acid by yeast (*—statistically different from the corresponding control variant α < 0.05).

**Figure 8 molecules-26-00806-f008:**
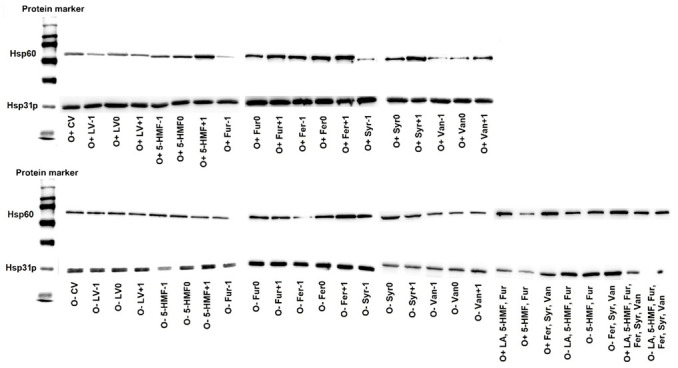
Effect of lignocellulose pretreatment by-products on the production of Hsp30p and Hsp60 by yeast (20 µg of protein per well).

**Table 1 molecules-26-00806-t001:** Concentration of Lignocellulosic Biomass Pretreatment By-Products in Experimental Variants Before and After *S. cerevisiae* Yeast Culture.

Research Variant	Concentration [mg/L] of By-Products Resulting from the Pretreatment of Lignocellulosic Biomass																	
	Levulinic Acid			5-HMF			Furfural			Ferulic Acid			Syringaldehyde			Vanillin		
	0 h	72 h	%_reduc._	0 h	72 h	%_reduc._	0 h	72 h	%_reduc._	0 h	72 h	%_reduc._	0 h	72 h	%_reduc._	0 h	72 h	%_reduc._
O^+^ CV	nf	nf	-	nf	nf	-	nf	nf	-	nf	nf	-	nf	nf	-	nf	nf	-
O^+^ LA^−1^	408.1 ± 7.9	233.0 ± 9.3	42.9 ± 3.1	nf	nf	-	nf	nf	-	nf	nf	-	nf	nf	-	nf	nf	-
O^+^ LA^0^	812.6 ± 23.8	565.1 ± 10.1	30.4 ± 2.0	nf	nf	-	nf	nf	-	nf	nf	-	nf	nf	-	nf	nf	-
O^+^ LA^+1^	1606.4 ± 29.2	1247.7 ± 25.9	22.3 ± 2.8	nf	nf	-	nf	nf	-	nf	nf	-	nf	nf	-	nf	nf	-
O^+^ 5-HMF^−1^	nf	nf	-	1344.4 ± 157.0	0.0 ± 0.0	100.0± 0.0	nf	nf	-	nf	nf	-	nf	nf	-	nf	nf	-
O^+^ 5-HMF^0^	nf	nf	-	2573.9 ± 82.1	0.0 ± 0.0	100.0± 0.0	nf	nf	-	nf	nf	-	nf	nf	-	nf	nf	-
O^+^ 5-HMF^+1^	nf	nf	-	5704.8 ± 249.3	0.0 ± 0.0	100.0± 0.0	nf	nf	-	nf	nf	-	nf	nf	-	nf	nf	-
O^+^ Fur^−1^	nf	nf	-	nf	nf	-	126.4 ± 9.7	0.0 ± 0.0	100.0± 0.0	nf	nf	-	nf	nf	-	nf	nf	-
O^+^ Fur^0^	nf	nf	-	nf	nf	-	545.6 ± 22.0	0.0 ± 0.0	100.0± 0.0	nf	nf	-	nf	nf	-	nf	nf	-
O^+^ Fur^+1^	nf	nf	-	nf	nf	-	1433.5 ± 40.7	0.0 ± 0.0	100.0± 0.0	nf	nf	-	nf	nf	-	nf	nf	-
O^+^ Fer^−1^	nf	nf	-	nf	nf	-	nf	nf	-	83.4 ± 0.9	0.0 ± 0.0	100.0± 0.0	nf	nf	-	nf	nf	-
O^+^ Fer^0^	nf	nf	-	nf	nf	-	nf	nf	-	176.0± 2.4	18.0 ± 1.5	89.8 ± 0.8	nf	nf	-	nf	nf	-
O^+^ Fer^+1^	nf	nf	-	nf	nf	-	nf	nf	-	376.3± 2.9	240.8 ± 26.7	36.0 ± 6.9	nf	nf	-	nf	nf	-
O^+^ Syr^−1^	nf	nf	-	nf	nf	-	nf	nf	-	nf	nf	-	19.5 ± 0.2	0.0 ± 0.0	100.0± 0.0	nf	nf	-
O^+^ Syr^0^	nf	nf	-	nf	nf	-	nf	nf	-	nf	nf	-	41.2 ± 0.9	0.0 ± 0.0	100.0± 0.0	nf	nf	-
O^+^ Syr^+1^	nf	nf	-	nf	nf	-	nf	nf	-	nf	nf	-	83.8 ± 0.4	0.0 ± 0.0	100.0± 0.0	nf	nf	-
O^+^ Van^−1^	nf	nf	-	nf	nf	-	nf	nf	-	nf	nf	-	nf	nf	-	19.4 ± 0.7	0.0 ± 0.0	100.0± 0.0
O^+^ Van^0^	nf	nf	-	nf	nf	-	nf	nf	-	nf	nf	-	nf	nf	-	39.1 ± 0.4	0.0 ± 0.0	100.0± 0.0
O^+^ Van^+1^	nf	nf	-	nf	nf	-	nf	nf	-	nf	nf	-	nf	nf	-	81.5 ± 0.5	0.0 ± 0.0	100.0± 0.0
O^−^ CV	nf	nf	-	nf	nf	-	nf	nf	-	nf	nf	-	nf	nf	-	nf	nf	-
O^−^ LA^−1^	588.7 ± 47.4	494.5 ± 40.2	16.0 ± 3.4	nf	nf	-	nf	nf	-	nf	nf	-	nf	nf	-	nf	nf	-
O^−^ LA^0^	863.8 ± 24.1	756.7 ± 31.6	12.4 ± 1.5	nf	nf	-	nf	nf	-	nf	nf	-	nf	nf	-	nf	nf	-
O^−^ LA^+1^	1709.1 ± 27.7	1595.5 ± 28.8	6.6 ± 1.8	nf	nf	-	nf	nf	-	nf	nf	-	nf	nf	-	nf	nf	-
O^−^ 5-HMF^−1^	nf	nf	-	1410.6 ± 39.6	0.0 ± 0.0	100.0± 0.0	nf	nf	-	nf	nf	-	nf	nf	-	nf	nf	-
O^−^ 5-HMF^0^	nf	nf	-	2676.5 ± 57.6	0.0 ± 0.0	100.0± 0.0	nf	nf	-	nf	nf	-	nf	nf	-	nf	nf	-
O^−^ 5-HMF^+1^	nf	nf	-	5386.3 ± 254.4	0.0 ± 0.0	100.0± 0.0	nf	nf	-	nf	nf	-	nf	nf	-	nf	nf	-
O^−^ Fur^−1^	nf	nf	-	nf	nf	-	128.6 ± 17.3	0.0 ± 0.0	100.0± 0.0	nf	nf	-	nf	nf	-	nf	nf	-
O^−^ Fur^0^	nf	nf	-	nf	nf	-	524.8 ± 75.5	0.0 ± 0.0	100.0± 0.0	nf	nf	-	nf	nf	-	nf	nf	-
O^−^ Fur^+1^	nf	nf	-	nf	nf	-	1350.9 ± 62.4	0.0 ± 0.0	100.0± 0.0	nf	nf	-	nf	nf	-	nf	nf	-
O^−^ Fer^−1^	nf	nf	-	nf	nf	-	nf	nf	-	108.0± 1.1	48.3 ± 7.9	55.3 ± 7.3	nf	nf	-	nf	nf	-
O^−^ Fer^0^	nf	nf	-	nf	nf	-	nf	nf	-	209.3± 1.7	114.8 ± 15.8	45.2 ± 7.1	nf	nf	-	nf	nf	-
O^−^ Fer^+1^	nf	nf	-	nf	nf	-	nf	nf	-	395.9± 5.4	307.7 ± 13.0	22.2 ± 3.7	nf	nf	-	nf	nf	-
O^−^ Syr^−1^	nf	nf	-	nf	nf	-	nf	nf	-	nf	nf	-	19.1 ± 1.2	0.0 ± 0.0	100.0± 0.0	nf	nf	-
O^−^ Syr^0^	nf	nf	-	nf	nf	-	nf	nf	-	nf	nf	-	40.6 ± 1.7	0.0 ± 0.0	100.0± 0.0	nf	nf	-
O^−^ Syr^+1^	nf	nf	-	nf	nf	-	nf	nf	-	nf	nf	-	85.6 ± 2.2	0.0 ± 0.0	100.0± 0.0	nf	nf	-
O^−^ Van^−1^	nf	nf	-	nf	nf	-	nf	nf	-	nf	nf	-	nf	nf	-	18.1 ± 0.6	0.0 ± 0.0	100.0± 0.0
O^−^ Van^0^	nf	nf	-	nf	nf	-	nf	nf	-	nf	nf	-	nf	nf	-	39.1 ± 1.0	0.0 ± 0.0	100.0± 0.0
O^−^ Van^+1^	nf	nf	-	nf	nf	-	nf	nf	-	nf	nf	-	nf	nf	-	83.3 ± 1.3	0.0 ± 0.0	100.0± 0.0
O^+^ LA, 5-HMF, Fur	822.3 ± 9.8	752.2 ± 27.2	8.5 ± 4.4	2727.5 ± 17.2	849.2 ± 16.6	68.9 ± 0.8	482.6 ± 30.0	0.0 ± 0.0	100.0± 0.0	nf	nf	-	nf	nf	-	nf	nf	-
O^+^ 5-HMF, Fur	nf	nf	-	2785.5 ± 38.5	143.4 ± 1.5	94.9 ± 0.1	479.4 ± 10.1	0.0 ± 0.0	100.0± 0.0	nf	nf	-	nf	nf	-	nf	nf	-
O^+^ Fer, Syr, Van	nf	nf	-	nf	nf	-	nf	nf	-	201.6± 2.6	114.9 ± 12.2	43.0 ± 6.4	39.5 ± 1.3	0.0 ± 0.0	100.0± 0.0	39.5 ± 1.3	0.0 ± 0.0	100.0± 0.0
O^−^ LA, 5-HMF, Fur	807.2 ± 3.3	733.9 ± 43.9	9.1 ± 5.4	2719.7 ± 56.6	0.0 ± 0.0	100.0± 0.0	509.2 ± 19.0	0.0 ± 0.0	100.0± 0.0	nf	nf	-	nf	nf	-	nf	nf	-
O^−^ 5-HMF, Fur	nf	nf	-	2692.8 ± 53.8	0.0 ± 0.0	100.0± 0.0	503.8 ± 20.1	0.0 ± 0.0	100.0± 0.0	nf	nf	-	nf	nf	-	nf	nf	-
O^−^ Fer, Syr, Van	nf	nf	-	nf	nf	-	nf	nf	-	201.6± 1.9	113.2 ± 14.0	43.9 ± 6.9	38.0 ± 0.6	0.0 ± 0.0	100.0± 0.0	39.5 ± 0.7	0.0 ± 0.0	100.0± 0.0
O^+^ LA, 5-HMF, Fur, Fer, Syr, Van	798.7 ± 14.2	741.3 ± 23.3	7.2 ± 3.8	2650.8 ± 90.0	489.5 ± 21.5	81.5 ± 1.3	459.4 ± 22.2	0.0 ± 0.0	100.0± 0.0	209.0± 4.3	173.2 ± 9.2	17.1 ± 4.5	41.7 ± 0.8	0.0 ± 0.0	100.0± 0.0	40.4 ± 1.5	0.0 ± 0.0	100.0± 0.0
O^−^ LA, 5-HMF, Fur, Fer, Syr, Van	769.1 ± 13.2	680.1 ± 14.1	11.5 ± 2.9	2705.1 ± 4.5	0.0 ± 0.0	100.0± 0.0	431.1 ± 19.6	0.0 ± 0.0	100.0± 0.0	208.3± 3.8	156.9 ± 15.1	24.6 ± 7.9	41.7 ± 0.7	0.0 ± 0.0	100.0± 0.0	39.7 ± 1.4	0.0 ± 0.0	100.0± 0.0

>nf—not found, O^−^—alcoholic fermentation, O^+^—aerobic conditions, CV—control variant without the stressors addition, LA—levulinic acid, 5-HMF—5-hydroxymethylfurfural, Fur—furfural, Fer—ferulic acid, Syr—syringaldehyde, Van—vanillin, superscript next to the experimental variant represents the stressor concentration level (^−1^—the lowest level; ^0^—the intermediate level; ^+1^—the highest level).

## Data Availability

Data sharing not applicable.

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
