# Peer review of "Impact of Lignocellulose Pretreatment By-Products on S. cerevisiae Strain Ethanol Red Metabolism during Aerobic and An-aerobic Growth"

_molecules, 2021, doi:10.3390/molecules26040806_

Round 1
Reviewer 1 Report
Molecules -1097780_ Metabolic reaction of S. cerevisiae Yeast to stress caused by the presence of By-products of lignocellulosic pretreatment in aerobic culture and during Alcoholic fermentation.
This manuscript highlights the growth of Ethanol Red in the presence of lignocellulosic inhibitors in both aerobic and anaerobic cultures. In addition to the growth studies, a ergosterol and trehalose were quantified and the expression of two heat shock proteins investigated. The growth studies are not all that novel and have been presented in numerous papers. The ergosterol and trehalose measurements are presented more thoroughly than other works and the heat shock proteins appears to be the most novel aspect of the work. Additional comments are listed below.
- With respect to the Western blots for detection of the heat shock proteins, while somewhat unsatisfying that densitometry not performed for a more quantitative view, as presented is fine. However, the amount of protein loaded in each well should be noted either in the figure legend (preferably) or the text. The reader should not have to go to the methods to find this information when authors make argument in text for a somewhat quantitative view of the results, page 14, lines 350-352.
- The title is a bit much and should be shortened. Title could simply be Impact of lignocellulosic inhibitors on S. cerevisiae strain Ethanol Red during aerobic and anaerobic growth. Not suggesting this be the title, but something to this effect that is more to the point and much shorter.
- The most novel aspect of the work is the blot of the heat shock proteins, but then seemingly the least amount of discussion on this topic. Expand the discussion a bit more on this and the impact of increased expression. Also, why these two proteins and not some of the others mentioned and what do authors expect to gain by measuring the expression of the other heat shock proteins. Is this a generalized stress response, under what other conditions is expression increased? Heat obviously, but any others.
- In a few places, the word analyzes is used when should be analysis, see lines 145, 345, and 423 for example. Check the manuscript for other instances.
Author Response
We would like to thank the Editor and the Reviewers for their comments and suggestions. We have revised the manuscript point by point according to the Reviewer’s comments. All the suggested changes are marked in yellow in the revised text and described below.
We hope that the quality and readability of our manuscript have been improved.
Response to Reviewer No. 1 comments:
This manuscript highlights the growth of Ethanol Red in the presence of lignocellulosic inhibitors in both aerobic and anaerobic cultures. In addition to the growth studies, a ergosterol and trehalose were quantified and the expression of two heat shock proteins investigated. The growth studies are not all that novel and have been presented in numerous papers. The ergosterol and trehalose measurements are presented more thoroughly than other works and the heat shock proteins appears to be the most novel aspect of the work. Additional comments are listed below.
With respect to the Western blots for detection of the heat shock proteins, while somewhat unsatisfying that densitometry not performed for a more quantitative view, as presented is fine. However, the amount of protein loaded in each well should be noted either in the figure legend (preferably) or the text. The reader should not have to go to the methods to find this information when authors make argument in text for a somewhat quantitative view of the results, page 14, lines 350-352.
As suggested by the Reviewer, additional information on the amount of protein applied to a single well was placed in the Figure 8 caption (in line 473).
The title is a bit much and should be shortened. Title could simply be Impact of lignocellulosic inhibitors on S. cerevisiae strain Ethanol Red during aerobic and anaerobic growth. Not suggesting this be the title, but something to this effect that is more to the point and much shorter.
The title has been shortened as suggested by the Reviewer and currently reads: “Impact of lignocellulose pretreatment by-products on S. cerevisiae strain Ethanol Red metabolism during aerobic and anaerobic growth”
The most novel aspect of the work is the blot of the heat shock proteins, but then seemingly the least amount of discussion on this topic. Expand the discussion a bit more on this and the impact of increased expression. Also, why these two proteins and not some of the others mentioned and what do authors expect to gain by measuring the expression of the other heat shock proteins. Is this a generalized stress response, under what other conditions is expression increased? Heat obviously, but any others.
The discussion section has been improved. We have added information on the role and occurrence of HSP proteins (lines 552-574).
In a few places, the word analyzes is used when should be analysis, see lines 145, 345, and 423 for example. Check the manuscript for other instances.
The text has been checked and corrected as suggested by the Reviewer (lines 144, 453, 531).
Reviewer 2 Report
The paper called “Metabolic reaction of S. cerevisiae yeast to stress caused by the presence of by-products of lignocellulose pretreatment in aerobic culture and during alcoholic fermentation” by KÅ‚osowski and Mikulski is an interesting analysis of the stress response of yeast caused by the product derived from lignocellulose hydrolysis. Those effects are studied in both respiratory and fermentative conditions, so this give plenty of information on the impact of some toxic compounds regarding the metabolic status of the cell. Besides interesting metabolic parameters such glycerol, ergosterol and trehalose production are studied together, so lots of information can be extracted.
However, the experiments are sometimes hard to follow. Graphs should indicate when the differences of the treatment compared to the control situation are statistically significant, with * or some other indication. Panels B, with the mixed compounds are confusing, the control is not included, nor there is line of its level, and there is not the same separation of O- and O+ conditions.
Minor points
- cerevisae should be italized in the title, keywords and throughout the text. In the abstract, line 12, indicate full name Saccharomyces cerevisiae.
Line 62. This line is repetitive.
Line 76. GRE2 and all genes in italics too.
Line 130. Are the authors talking about heat shock proteins or total protein in this case?
Line 151. Use “ethanol” instead of “ethyl alcohol”.
Line 163. The superscripts to indicate the levels of chemical used are confusing, as it would increase by ten-fold. Is "0" representing an average?
Figure 8. Indicate the different gels used in the figure with frames. Is there any loading control, a coomassie staining?
The Discussion section has many information that would fit in Introduction and it lacks more speculation on the molecular causes behind the phenotypes.
Author Response
We would like to thank the Editor and the Reviewers for their comments and suggestions. We have revised the manuscript point by point according to the Reviewer’s comments. All the suggested changes are marked in yellow in the revised text and described below.
We hope that the quality and readability of our manuscript have been improved.
Response to Reviewer No. 2 comments:
The paper called “Metabolic reaction of S. cerevisiae yeast to stress caused by the presence of by-products of lignocellulose pretreatment in aerobic culture and during alcoholic fermentation” by KÅ‚osowski and Mikulski is an interesting analysis of the stress response of yeast caused by the product derived from lignocellulose hydrolysis. Those effects are studied in both respiratory and fermentative conditions, so this give plenty of information on the impact of some toxic compounds regarding the metabolic status of the cell. Besides interesting metabolic parameters such glycerol, ergosterol and trehalose production are studied together, so lots of information can be extracted.
However, the experiments are sometimes hard to follow. Graphs should indicate when the differences of the treatment compared to the control situation are statistically significant, with * or some other indication. Panels B, with the mixed compounds are confusing, the control is not included, nor there is line of its level, and there is not the same separation of O- and O+ conditions.
As suggested by the Reviewer, the Figures have been corrected and the arrangement of the experimental variants has been changed according to the culture conditions. Statistical analysis has been added in lines 233, 271, 310, 356, 409, 444.
Minor points:
cerevisae should be italized in the title, keywords and throughout the text. In the abstract, line 12, indicate full name Saccharomyces cerevisiae.
Corrected (lines 12, 125, 127, 164, 457, 494, 499, 501, 503, 504, 509, 512, 524, 537, 577, 591, 602, 664, 698).
Line 62. This line is repetitive.
Duplicate content has been removed.
Line 76. GRE2 and all genes in italics too.
Corrected (in lines 75, 519,524).
Line 130. Are the authors talking about heat shock proteins or total protein in this case?
Corrected to "total protein" (line 129).
Line 151. Use “ethanol” instead of “ethyl alcohol”.
Done (lines 150, 153).
Line 163. The superscripts to indicate the levels of chemical used are confusing, as it would increase by ten-fold. Is "0" representing an average?
Superscripts informing about the level of contamination of a given culture medium (according to Table 1) are used to simplify the notation. The superscripts "-1, 0, +1" represent successive levels of stressor concentration (from the lowest to the highest). The superscript "0" represents the intermediate level of the stressor added to the culture medium according to Table 1. As suggested by the Reviewer, an appropriate explanation has been added (lines 161-162).
Figure 8. Indicate the different gels used in the figure with frames. Is there any loading control, a coomassie staining?
The study focused on the analysis of HSP by Western Blot method, and not on the analysis of the protein profile after staining the gels with Coomassie blue. A chemiluminescent molecular weight standard was used to visualize individual proteins. During the incubation with the antibodies, an HRP tag dedicated to the protein standard was added. As suggested by the Reviewer, additional information has been provided in Figure 8.
The Discussion section has many information that would fit in Introduction and it lacks more speculation on the molecular causes behind the phenotypes.
The Discussion section has been improved as suggested by the Reviewer (lines 542-546, 552-574).